# Exploring the Potential Medicinal Benefits of *Ganoderma lucidum*: From Metabolic Disorders to Coronavirus Infections

**DOI:** 10.3390/foods12071512

**Published:** 2023-04-03

**Authors:** Elif Ekiz, Emel Oz, A. M. Abd El-Aty, Charalampos Proestos, Charles Brennan, Maomao Zeng, Igor Tomasevic, Tahra Elobeid, Kenan Çadırcı, Muharrem Bayrak, Fatih Oz

**Affiliations:** 1Department of Food Engineering, Agriculture Faculty, Ataturk University, Erzurum 25240, Türkiye; 2Department of Pharmacology, Faculty of Veterinary Medicine, Cairo University, Giza 12211, Egypt; 3Department of Medical Pharmacology, Medical Faculty, Ataturk University, Erzurum 25240, Türkiye; 4Laboratory of Food Chemistry, Department of Chemistry, School of Sciences, National and Kapodistrian University of Athens Zografou, 15784 Athens, Greece; 5School of Science, RMIT University, Melbourne, VIC 3001, Australia; 6Riddet Institute, Palmerston North 4442, New Zealand; 7State Key Laboratory of Food Science and Technology, Jiangnan University, Wuxi 214122, China; 8International Joint Laboratory on Food Safety, Jiangnan University, Wuxi 214122, China; 9Faculty of Agriculture, University of Belgrade, 11000 Belgrade, Serbia; 10The German Institute of Food Technologies (DIL) Professor-von-Klitzing-Straße 7, 49610 Quakenbrück, Germany; 11Human Nutrition Department, College of Health Sciences, QU Health, Qatar University, Doha P.O. Box 2713, Qatar; 12Department of Internal Medicine, Erzurum Regional Training and Research Hospital, Health Sciences University, Erzurum 25240, Türkiye

**Keywords:** *Ganoderma lucidum*, bioactive compounds, COVID-19, antioxidant, anticancer, cardiovascular diseases, antidiabetic

## Abstract

*Ganoderma lucidum* is a medicinal mushroom that has been traditionally used in Chinese medicine for centuries. It has been found to have a wide range of medicinal properties, including antioxidant, anti-inflammatory, and immune-boosting effects. Recent research has focused on the potential benefits of *G. lucidum* in treating metabolic disorders such as diabetes and obesity, as well as its possible role in preventing and treating infections caused by the coronavirus. Triterpenoids are a major group of bioactive compounds found in *G. lucidum*, and they have a range of biological activities, including anti-inflammatory and antioxidant properties. These compounds have been found to improve insulin sensitivity and lower blood sugar levels in animal models of diabetes. Additionally, *G. lucidum* polysaccharides have been found to reduce bodyweight and improve glucose metabolism in animal models of obesity. These polysaccharides can also help to increase the activity of certain white blood cells, which play a critical role in the body’s immune response. For coronavirus, some in vitro studies have shown that *G. lucidum* polysaccharides and triterpenoids have the potential to inhibit coronavirus infection; however, these results have not been validated through clinical trials. Therefore, it would be premature to draw any definitive conclusions about the effectiveness of *G. lucidum* in preventing or treating coronavirus infections in humans.

## 1. Introduction

There is a growing trend among consumers to seek out foods and dietary supplements that not only provide basic nutritional needs but also offer additional health benefits that may promote longevity and overall well-being. Functional foods are an important part of the growing trend toward personalized nutrition and a more holistic approach to health and wellness. Functional foods are able to produce pharmacological effects because of the bioactive compounds they contain. The concept of using foods for their therapeutic properties dates back thousands of years and is a fundamental principle in traditional medicine systems. In this context, mushrooms have been used for their medicinal properties in traditional medicine systems for centuries, and scientific research has confirmed many of their therapeutic benefits [1].

For many years, edible mushrooms have been favored by consumers for their distinct flavor and aroma and have also been utilized for their medicinal properties. In fact, mushrooms are not only commonly found in kitchens but are also used as components of traditional herbal medicines due to the beneficial substances they contain [2]. *Ganoderma* is a macrofungus that is often used for its medicinal and functional food properties. This multiporous mushroom typically grows on the stumps of various deciduous trees, including oak, maple, elm, willow, sweetgum, magnolia, and acacia [3,4,5]. In addition, it also grows in Europe, Asia, and North and South America, especially in temperate regions [5]. In contrast, planting media, such as paddy husk, brown rice flour, and rubber tree wastes, are favored for cultivating *Ganoderma* without the use of chemicals, pesticides, or hormones. To ensure the preservation of the mushroom’s therapeutic properties, it is recommended to carefully regulate the use of sunlight during cultivation [6].

*Ganoderma lucidum*, a species of mushroom in the *Ganoderma* genus of the Ganodermataceae family, is the most extensively studied and well known. This mushroom is known by different names across various countries and cultures. In Japan, it is referred to as “Reishi”, which means “spiritual potency”. In China, it is known as “Lingzhi”, which means “divine mushroom”, while in Korea, it is called “Youngzhi”, meaning “mushroom of immortality”. Traditional medicine practices in these countries have utilized *G. lucidum* for many years, with beliefs in its various health benefits. In contrast, research on these mushrooms in Western studies dates back only to the past 30 years [5,6,7,8]. Based on the taxonomy information of *G. lucidum*, it belongs to Kingdom: Fungi, Phylum: Basidiomycota, Class: Basidiomycetes, Order: Polyporales, Family: Ganodermataceae, Genus: *Ganoderma*, and Species: *Ganoderma lucidum* P. Karst [6,9]. However, there is currently ongoing debate within the scientific community about the correct scientific name for this species. Some authors have proposed alternative names such as *G. lingzhi*, *G. reishi*, *G. mannentake*, or *G. lingzhi var. lucidum* [5].

*G. lucidum* was recorded in the ancient Chinese herbal encyclopedia, Shen-Nong-Ben-Cao-Jing [10]. The genus *Ganoderma* was established by the Italian mycologist Carlo Vittadini in 1831 based on his study of various species of fungi, including Ganoderma, and *G. lucidum* was the first species in the genus to be described [11]. While there are over 250 species of *Ganoderma*, 6 species are among the most commonly used: *G. lucidum*, *G. applanatum*, *G. sinense*, *G. tsugae*, *G. capense*, and *G. boinense* [5,11] (Figure 1).

The growth of *G. lucidum* occurs in limited quantities in nature, and it takes several months to complete the fruiting process [12]. The structure of *G. lucidum* is generally fan-shaped, kidney-shaped, or semicircular, and it typically exhibits a dark red, reddish-brown, or reddish-black color. Yellow or ochre hues become more prominent around the edges. The mushroom flesh is typically yellowish brown to dark brown in color [13]. The Latin root, “Lucidum”, meaning bright, has been linked to the shiny appearance of these mushrooms [5,13]. Table 1 provides details on the various colors and applications of *G. lucidum* [3].

An analysis of *G. lucidum* extract revealed that it consists of approximately 11.1% glucose, 10.2% minerals (including K, Mg, and Ca), and 7.3% protein [14]. However, *G. lucidum* also contains a variety of biologically active components, such as polysaccharides, triterpenes, flavonoids, alkaloids, steroids, unsaturated fatty acids, proteins, amino acids, enzymes, vitamins, and minerals. Among these, the therapeutic properties of *G. lucidum* have primarily been attributed to its polysaccharides and triterpenoids, which have been the focus of many studies [10,14,15].

In general, the components of *G. lucidum* can be listed as follows:Polysaccharides and glycoproteins (α-D-glucans, β-glucans, β-D-glucans, polysaccharide–protein complexes, and α-D-mannans) [5,15,16].Terpenoids (ganoderic acid A–Z, ganoderals, ganosporeric acid A, lucidenic acids, and ganoderiols) [5,15,16].Nitrogenous compounds (aspartic acid, glutamic acid, lysine, leucine, methionine, cysteine, and nucleotides) [16].Other components (phosphorus, germanium, calcium, sulfur, and magnesium) [16].

Studies have found that *G. lucidum* contains over 432 secondary metabolites, as well as more than 200 different types of polysaccharides. Among the secondary metabolites, there are over 380 terpenoids and more than 30 steroidal groups [15]. The types and quantities of the bioactive compounds in *G. lucidum* can vary depending on factors such as the place of production, growing conditions, methods of compound extraction, and strain selection [17]. Figure 2 outlines the various techniques that have been utilized for the extraction of triterpenoids and polysaccharides from *G. lucidum*.

*G. sinense* is another species of the *Ganoderma* genus and is easily distinguished by its purple color. It contains a range of bioactive compounds, including polysaccharides, ergosterol, coumarin, organic acids, glucosamine, mannitol, polysaccharide alcohol, fatty acids, alkaloids, water-soluble proteins, and enzymes. These compounds have been shown to possess various therapeutic properties, such as antitumor, anticytopenia, immune stabilization, antioxidant, and mushroom poison detoxification activities [7]. Another species within the *Ganoderma* genus, *G. austral*, is utilized similarly to other species in traditional medicine for treating various diseases. Research has shown that *G. austral* possesses neuroprotective properties, as well as cytotoxic and α-glucosidase inhibition activities [19]. Research has indicated that *G. tsugae* displays anti-inflammatory and anticancer properties, while *G. applanatum* exhibits antifibrotic effects, in addition to the aforementioned activities [11].

### 1.1. Ganoderma Lucidum Components

#### 1.1.1. Triterpenes

Triterpenes are a type of organic compound derived from isoprene units and belong to the larger group of naturally occurring compounds called terpenes. With a C30 skeleton structure, triterpenes typically have molecular weights ranging from 400 to 600 g/mol [1]. Terpenoids can be categorized into different types, such as volatile triterpenoids and sterols, essential oils, less volatile diterpenes, and carotenoids [17]. According to reports, the bitterness of Ganoderma’s fruiting body is due to its triterpene/triterpenoid secondary metabolites. It is worth noting that an increase in bitterness results in an increase in triterpene content [6,9]. *G. lucidum* contains various triterpenoids, including C30 lanostans, such as ganoderic acids, aldehydes, alcohols, esters, glycosides, lactones, and ketones. Additionally, it contains C30 pentacyclic triterpenes, C24 lanostans, C27 lanostans (including lucidenic acids, alcohols, lactones, and esters), and C25 lanostans [1,6,13,20,21,22]. Furthermore, it was found that, among these triterpenoids, ganoderic acid DM, ganoderic acid T, ganoderic acid C, ganoderic acid H, ganoderic acid A, ganoderic acid F, ganoderic acid X, and ganoderic acid Y exhibited the most potent bioactivity [8,10]. Triterpenes possess various beneficial activities, such as antitumor, anti-inflammatory, antioxidant, antihepatitis, antimalarial, hypoglycemic, antimicrobial, and anti-HIV activities [20].

#### 1.1.2. Polysaccharides

Polysaccharides in *Ganoderma* are synthesized using different types of monosaccharides, such as glucose, fructose, xylose, mannose, and galactose. The various glycosidic bonds between these monosaccharides lead to the formation of diverse polysaccharides that exhibit unique properties and potential health benefits [4,20]. *G. lucidum* contains multiple bioactive polysaccharides, including (1–3)-α/β-glucans, (1–6)-α/β-glucans, α-D-mannans, glycoproteins, and heteropolysaccharides, that are soluble in water [4,13]. Among these polysaccharides, β-1-3 and β-1-6-D-glucans are particularly noteworthy for their bioactive properties [8,12,15]. Research has confirmed that the polysaccharides found in *G. lucidum* exhibit several beneficial effects, such as anticancer, antioxidant, antimicrobial, immunomodulatory, hypoglycemic, and anti-inflammatory effects, and also protect the intestinal mucosal barrier mechanism [15,20,23].

#### 1.1.3. Other Bioactive Components

*Ganoderma* contains high levels of germanium, which possesses antimutagenic, antitumor, immune-enhancing, and antioxidant properties [5]. Studies have found that certain proteins extracted from *Ganoderma* fruit and spores, namely, LZP-1, LZP-2, and LZP-3, exhibit mitogenic activity. In addition, the Zhi-8 protein isolated from micelles shows immunomodulatory activity. Furthermore, bioactive peptides such as lectins, ribonucleases, and laccases have been identified in *Ganoderma* [6,18]. *Ganoderma* is a rich source of enzymes, such as superoxide dismutase, lysozyme, and protein enzymes, that are essential for our body’s disease defense and metabolic processes. Research has shown that *Ganoderma’s* steroid, ganodosterone, possesses antihepatotoxic activity. Furthermore, *Ganoderma* is a source of vitamins C and E as well as β-carotene [6].

### 1.2. Consumption of G. lucidum and Its Effects on Health

Due to their tough and rigid fruiting bodies, *Ganoderma* species are not considered suitable for consumption as culinary mushrooms; however, they are highly valued as medicinal mushrooms, and their extracts are utilized as remedies for treating various ailments [9]. Mushrooms are commonly consumed in various forms, such as capsules, liquid extracts, chewable tablets, rice wine, tea, syrup, hair and skin care products, and tea, to obtain their therapeutic benefits. *G. lucidum*, for instance, is dried, powdered, and marketed as a dietary supplement to harness its pharmacological properties. The global sales of these products are estimated to exceed US$ 2.5 billion annually [9,11,24,25]. Research has established that *G. lucidum* exhibits a diverse range of beneficial effects, including, but not limited, to anticancer (in prostate, lung, breast, and colon), antidiabetic, anti-inflammatory, antioxidant, antihepatitis, immunomodulatory, hypocholesterolemic, antimicrobial, hypoglycemic, cardioprotective, antihyperpigmentation, antiarthritic, proapoptotic, antiallergic, antianxiety, antiandrogenic, and antinociceptive effects [1,8,15,19]. The health benefits associated with *G. lucidum* are attributed to its spores, micelles, and bioactive compounds extracted from the fruiting body [8,23]. Figure 3 [4,8,10,15,23,26,27,28] illustrates the therapeutic effects of diverse bioactive compounds extracted from *G. lucidum*. Moreover, this article provides an in-depth discussion of the mechanisms of action of these compounds concerning COVID-19, antioxidants, cardiovascular diseases, and antidiabetes and anticancer effects.

Some researchers suggest that *Ganoderma* is not only beneficial but also safe for consumption, as there have been no reports of toxicity associated with its general use [6]. However, a study reported that patients experienced transient symptoms such as thirst, drowsiness, redness, bloating, frequent urination, abnormal sweating, and diarrhea when given oral doses of 1.5–9 g/day of Ganoderma extract powder [3]. Studies have also indicated that, depending on the dose and individual health conditions, *Ganoderma* consumption may lead to various disorders, such as allergies, blood clots, and liver problems [27].

#### 1.2.1. Mechanism of Action as Antioxidant

Chain reactions, resulting in various forms of damage and loss, occur in both our bodies and food due to oxidation reactions. In food, oxidation is the process by which unsaturated fatty acids react with oxygen, converting them into low molecular weight compounds. Within the body, the process begins with breathing and is initiated by the formation of free radicals during metabolic events. These reactive oxygen species (ROS) can adversely affect cells, causing disruptions to normal bodily functions. ROS, similarly to lipid oxidation products, have been linked to various ailments, such as cancer, cardiovascular diseases, neurodegenerative diseases, diabetes, inflammatory diseases, immune system disorders, and aging [29,30,31].

Oxidation is responsible for diminishing the nutritional content of foods, generating harmful compounds, and compromising the product’s shelf life and appeal to consumers. To minimize the impact of oxidation, various techniques are implemented to eliminate pro-oxidants and prevent air exposure. However, in practice, it is difficult to achieve complete prevention of oxidation, and, therefore, antioxidants are often added to food processing. Antioxidants function by scavenging free radicals, chelating pro-oxidative metals, extinguishing singlet oxygen and photosensitive substances, and inactivating lipoxygenase, effectively slowing down food oxidation [31,32,33,34]. Phenolic compounds such as tocopherols, flavonoids, phenolic acids, carotenoids, amino acids, and ascorbic acid are commonly employed as antioxidant agents to impede the oxidation process by reacting with free radicals. Metal chelation, on the other hand, involves the participation of various components, such as phospholipids, polyphenols, amino acids (tryptophan, methionine, and cysteine), and peptides. In addition to their effects on food oxidation, antioxidant compounds present in foods can also prevent oxidation reactions from occurring within the body when consumed; however, the efficacy of these compounds within the body can vary based on factors such as concentration, the form in which they are present, and their degree of absorption [32].

The antioxidant properties of *G. lucidum* have been scientifically established and are attributed to its constituent polysaccharides, glycoproteins, triterpenes, amino acids, and phenolic compounds. Polysaccharides in *G. lucidum* exhibit antioxidant effects via various mechanisms, such as terminating or preventing chain reactions by donating hydrogen, donating electrons, or combining hydrogen and electrons with free radicals. Additionally, *G. lucidum* polysaccharides have been shown to reduce lipid peroxidation and malondialdehyde, an oxidative product, and positively impact the activity of glutathione peroxidase, the primary defense mechanism against free radicals in the body’s cells. This effect is achieved by stimulating the synthesis of catalase and superoxide dismutase enzymes via a redox cycle [4,22]. Furthermore, the antioxidant effects of triterpenes and leucine amino acids, as well as glycoproteins produced by combining polysaccharides with peptides or proteins through covalent bonds, have also been well established [4,5]. The ethanolic extracts of *G. lucidum* have been found to contain various phenolic compounds, including quercetin, myricetin, gallic acid, chlorogenic acid, protocatechin, cinnamic acid, *p*-hydroxybenzoic acid, and *p*-coumaric acid, which have exhibited antioxidant activities [20,35,36]. Taofiq et al. [20] utilized the Soxhlet system to extract an ethanolic extract from the fruiting roots of *G. lucidum* and analyzed its antioxidant activity and chemical composition. Their findings revealed high concentrations of ganoderic acid A, C2, and H, as well as several phenolic acids, such as *p*-hydroxybenzoic, protocatechuic, and syringic acid. Interestingly, the phenolic acids present in the extract exhibited greater efficacy in scavenging DPPH radicals compared to the polysaccharides, indicating that the extract’s antioxidant activity may largely stem from the phenolic acid content. In addition, Veljović et al. [36] conducted an analysis on the ethanolic extracts of *G. lucidum* to determine their chemical composition and antioxidant activity. The extracts were found to contain significant amounts of hesperetin and naringenin, as well as varying levels of quercetin, gallic acid, kaempferol, and trans-cinnamic acid. The researchers also reported that the extracts displayed noteworthy levels of antioxidant activity; however, they observed a decline in antioxidant activity due to interactions between the polysaccharides and phenolic compounds present in the extracts. This indicates that the antioxidant activity of the extracts may be impacted by the interactions between different chemical constituents. Despite these findings, it has been noted that there is inadequate research conducted on the antioxidant capabilities of the bioactive compounds present in *G. lucidum*, and further extensive studies are required for their identification [5].

According to Ryu et al. [37], freeze-dried *G. lucidum* was found to possess superior antioxidant activity in comparison to heat-dried *G. lucidum*. The study further revealed that the drying time and the levels of polysaccharides, lucidenic acid, and 12-acetyl ganoderic acid F were key factors contributing to the highest antioxidant activity observed in the extracts.

The antioxidant activity of *G. lucidum* spore water extract was examined by Zhang et al. [38] using 1,1-diphenyl-2-picrylhydrazyl (DPPH) and 2,2′-azino-bis (3-ethylbenzothiazoline-6-sulfonic acid) diammonium salt (ABTS) analyses. The results of the study indicated that the DPPH and ABTS values of the extracts were 0.2456 and 0.3637 mg/mL, respectively. The researchers attributed the high antioxidant activity to the ability of polysaccharides in *G. lucidum* to donate hydrogen atoms and convert free radicals into stable structures, and, thus, breaking the chain reactions of free radicals.

#### 1.2.2. Mechanism of Action as Anticancer

The uncontrolled proliferation of abnormal cells in any part of the body that affect healthy cells and can spread to other tissues is known as cancer, the primary cause of global mortality [39,40,41]. Cancer cells divide uncontrollably, forming a mass that can metastasize and grow uncontrollably in other organs through the blood and lymphatic systems. Cancer is an entirely unregulated process influenced by a variety of factors, including living conditions, such as nutrition, smoking, and alcohol consumption; environmental toxins; the immune system; genetic predisposition; radiation; chemical exposure; and changes in formation and progression [39]. One of the significant factors that have a considerable impact on the development of cancer is nutrition, which is one of the primary focuses for prevention and has been extensively studied [42,43]. According to the American Institute for Cancer Research (AICR) and the World Cancer Research Fund (WCRF), maintaining appropriate nutrition, exercise, and bodyweight can reduce the risk of developing all types of cancer by 30–40%. To prevent or treat cancer, a diet should contain essential components such as selenium, vitamins (especially folic acid, vitamins B12, C, D, and E), a variety of antioxidants (such as α-carotene, β-carotene, lycopene, lutein, and cryptoxanthin), and probiotics [42,44].

Cancer treatment involves a range of approaches, including surgery, chemotherapy, radiation therapy, photodynamic therapy, thermal therapy, immunotherapy, and gene therapy, either individually or in combination; however, due to the side effects of some treatments and their harmful impact on healthy cells, the quest for natural techniques and substances in cancer treatment has intensified. As a result, researchers have started exploring the anticancer activity of several food components to determine their effectiveness against cancerous cells [43,45]. Several components, including phenolic compounds, flavonoids, and terpenoids, have demonstrated anticancer activity through their various functions. Phenolic compounds inhibit the invasion and metastasis of cancer cells. Flavonoids fight angiogenesis, increase DNA fragmentation, inhibit signal transduction enzymes, and promote the phosphorylation of the epidermal growth factor receptor. Terpenoids, on the other hand, inhibit signal transduction of antiapoptotic proteins, activate proapoptotic mediators, and induce cell cycle arrest, contributing to the formation and prevention of cancer [43]. *G. lucidum*-derived polysaccharides and triterpenoids have also been demonstrated to exhibit anticancer activity through various mechanisms. The polysaccharides from *G. lucidum* act by stimulating the immune system (macrophages, T cells, and B cells) to produce cytokines and activate the anticancer activities of immune cells [14,17]. Apart from bolstering the immune system, polysaccharides exhibit anticancer activity via several mechanisms, including inducing a cytotoxic effect, reducing integrin expression to hinder tumor cell adhesion, promoting apoptosis of cancer cells, and impeding angiogenesis [15,23,46]. Polysaccharides with anticancer activity in *G. lucidum* have long glycosidic bonds and high molecular weights [14]. In this context, it has been stated that branched β-1-3- and β-1-6-D-glucans from polysaccharides show anticancer activity via complement receptor type 3, which binds β-glucan polysaccharides. Glucuronoglucan, glucogalactan, mannogalactoclucan, and arabinoglucan also show antitumor activity [15,47]. Ganoderan A-B-C and glycoproteins (heteropolysaccharides) are cited as examples of high molecular weight anticancer compounds [15]. It has been reported that *G. lucidum* triterpenoids show anticancer activity by inhibiting the metastatic growth of cancerous cells, suppressing the attack of cancerous cells, and inhibiting protein kinase C or β-catenin activity [17,27,37]. Triterpenoids, especially ganoderic acids, have been proven to have cytotoxicity in various cancer cells [47]. Among ganoderic acids, ganoderic acid T, ganoderic acid D, and ganoderiol F have been reported to have effective anticancer activity [27]. In addition, the cytotoxic effects of U, V, W, X, and Y ganoderic acids on hepatoma cells have also been proven [47]. Furthermore, *G. lucidum* contains valuable bioactive compounds with immunomodulatory and anticancer properties that can serve as natural resources to mitigate the toxicity of conventional chemotherapy and/or radiation and enhance the immune system of cancer patients [27]. Sui et al. [48] studied the immunomodulatory effect of polysaccharides isolated from *G. lucidum* on the typical macrophage cell line, RAW 246.7, and the inhibitory effect on the human carcinoma cell line, HepG2. Researchers determined that 1,4-α-glucans has a stronger inhibition of HepG2 cells than β-glucan isolated from the fruiting body. They also revealed that β-glucan has a stronger immunomodulating effect. In this study, it was reported that 1,4-α-glucans isolated from *G. lucidum* exhibited the broad-spectrum inhibition of cancer cells in a dose-dependent manner. Chen et al. [49] reported that triterpenoids isolated from *G. lucidum* exhibit cytotoxic activity on the human breast carcinoma cell line, MDA-MB-231, and hepatocellular carcinoma cell line, HepG2. Table 2 shows the anticancer mechanisms of compounds isolated from *G. lucidum* in detail.

#### 1.2.3. Mechanism of Action against Cardiovascular Diseases

Cardiovascular diseases (CVDs), which are among the leading causes of death worldwide, such as cancer, are related to the heart and vessels and can be classified as hypertensive heart disease, cardiomyopathy, coronary heart disease, and heart failure. The causes of these diseases are hypertension, atherosclerosis, dyslipidemia, inflammation, oxidative stress, and enteric dysbacteriosis [53,54,55,56]. According to a World Health Organization (WHO) report in 2019, 32% of deaths worldwide (approximately 17.9 million people) were caused by CVD [57]. On the other hand, it is estimated that this number may reach 22.2 million by 2030 [56]. Various factors, such as age, genetic factors, unhealthy diet, smoking, alcohol use, obesity, and lack of physical activity, influence the development of CVD [56,57]. Among the mentioned factors, nutrition is vital in preventing the formation of primary and secondary CVD. The drugs used to treat these diseases cause side effects in patients, and the positive effects of some natural compounds that have been observed in foods present them as an alternative for the treatment of CVD [55,56,58]. It has been reported that natural antioxidant compounds (such as polysaccharides, polyphenols, anthocyanins, gallate, epigallocatechin rutin, quercetin, and puerarin) contribute to the reduction in CVD risk through mechanisms such as regulating the lipid profile, reducing blood pressure, preventing oxidative stress, improving the gut microbiota, and reducing the occurrence of inflammation [55,59,60]. In recent years, it has been proven that functional foods containing components that play an active role in physiological functions have therapeutic properties for many diseases, such as CVD. It is thought that the protective ability of functional foods for heart health is due to their properties, such as cardioprotective activity, antioxidant effects, and the lowering of blood lipid levels [60].

It has been observed that *G. lucidum*, which has been used in traditional medicine for many years in the Far East and defined as the mushroom of immortality, has positive effects on cardiovascular diseases. In general, the positive effects of *G. lucidum* extracts on CVD are based on their effects on blood pressure, lipids, diabetes, and obesity; preventing blood platelets from sticking together; and reducing low-density lipoprotein cholesterol, total cholesterol, and triglycerides due to their antioxidant activities [1,27,61,62]. Indeed, in a study that investigated the effect of hydroalcoholic extracts from *G. lucidum* on hypocholesterolemic properties, it was determined that the application of 0.5–1% dose range extract in mice caused a reduction in serum cholesterol, low-density lipoprotein cholesterol (LDL-C), and triglyceride levels [61]. In this context, Wu et al. [62] reported that a dose of 50 mL/day for 1 month reduced LDL-C in volunteers who were fed *G. lucidum* mycelium-fermented liquid.

It has been determined that triterpenes, one of the important bioactive compounds of *G. lucidum*, play a preventive role in CVD with their inhibitory and anti-inflammatory activities and by scavenging cellular reactive oxygen species through their antioxidant activities. *G. lucidum* contains various bioactive compounds, such as polysaccharides and peptides, that may prevent CVD by protecting endothelial cells in blood vessels [27,63]. In addition, they have other cardiovascular benefits, such as lowering blood pressure and improving lipid metabolism [27]. Adeyi et al. [24] conducted a study on the effects of *G. lucidum* ethanol extract on metabolic syndrome in rats. The study investigated the potential hypoglycemic, hypolipidemic, hypotensive, and antioxidant properties of the ethanol extract in rats with induced metabolic syndrome. They found that a dose of 70 mg/kg bodyweight of *G. lucidum* ethanol extract exhibited antioxidant, antihypertensive, antihyperglycemic, and antidyslipidemic activities. Shaher et al. [64] investigated the effect of *G. lucidum* spores applied to mice on antidiabetic cardiomyopathy and reported that the application of *G. lucidum* spores in mice at a dose of 300 mg/kg for 70 days reduced blood glucose levels by 20.3% and triglyceride levels by 20.4%. In the same study, it was concluded that *G. lucidum* spore application alleviated diabetic cardiomyopathy by reducing hyperglycemia, oxidative stress, inflammation, and apoptosis.

#### 1.2.4. Antidiabetic Mechanism of Action

The term diabetes, which was first used by the Greek doctor Aretus in the 2nd century, was defined by the WHO in 1999 as a metabolic disorder resulting in disruptions in carbohydrate, fat, and protein metabolism due to insulin secretion or the effects of insulin [65]. Diabetes mellitus (DM) is a metabolic disease that was briefly defined as high blood sugar for a long time. Diabetes is associated with symptoms such as increased hunger/thirst and frequent urination [66]. Diabetes progresses when normal fasting blood sugar, which is in the range of 70–100 mg/dL, is constantly above these values. This change in blood sugar is due to insulin deficiency or obstructive factors in the tissues that incorporate insulin into metabolism [65]. Diabetes is seen in two main forms: Type 1 and Type 2. Type 1 is caused by insufficient insulin production due to the destruction of beta cells in the pancreas. Type 2 diabetes, on the other hand, begins with insulin resistance, which is the insufficiency of insulin secretion in the body and the inability of cells to respond to insulin correctly [66,67].

According to a report published by the IDF Diabetes Atlas in 2021, there are currently 537 million adults with diabetes between the ages of 20 and 79. This number is estimated to increase to 783 million by 2045 [68]. The WHO reports 422 million diabetic cases and 1.5 million deaths due to diabetes [69]. Approximately 90% of cases of diabetes, which is seen in very high numbers worldwide and is the sixth highest cause of death, have been determined to be Type 2 diabetes. On the other hand, in addition to reducing quality of life, diabetes has been reported to be a precursor to severe disorders, such as cardiovascular diseases, kidney diseases, cancer, and foot ulcers [67]. Therefore, the treatment of diabetes has gained even more importance. While there is no preventive measure for type 1 diabetes, treatment is applied for type 2 diabetes using insulin or noninsulin medication drugs. For this purpose, various antidiabetic drugs (α-glucosidase inhibitors, sulfonylureas, secretagogues, and biguanides) are frequently preferred in diabetes control. Over time, resistance to drugs or inhibitors used in the treatment, and side effects such as bloating, gas, meteorism, and diarrhea, have led to the search for other sources [66,70]. To eliminate these problems in treating diabetes, natural sources with α-amylase and α-glucosidase inhibitory activities have attracted the attention of researchers.

Compounds, such as glycosides, proteins, and lipids in foods, that inhibit α-amylase and α-glucosidase are promising, as they reduce the risk of any toxicity or side effects in the treatment of diabetes [70]. One option, containing bioactive compounds, with this feature is *G. lucidum*. The antidiabetic activities of polysaccharides, triterpenoids, proteins, and glycoproteins extracted from *G. lucidum* have been proven by various studies. While the antidiabetic mechanisms of polysaccharides, one of these bioactive compounds, were demonstrated by the expression of enzymes important to glucose metabolism (by increasing phosphofructokinase, glucose-6-phosphate dehydrogenase, and hepatic glucokinase, but inhibiting manganese superoxide dismutase, glycogen synthetase, and glutathione peroxidase), the activity of glycoproteins was explained by the inhibition of protein tyrosine phosphatase 1B. The antidiabetic mechanism of triterpenoids, another bioactive compound, is expressed by the inhibition of α-glucosidase with aldose reductase, the enzyme that converts glucose into sorbitol. Finally, it has been reported that Ling Zhi-8 protein can decrease plasma glucose concentration by decreasing lymphocyte infiltration and increasing insulin antibody detection, and that is also has antidiabetic activity [16,70]. Ganoderan A, B, and C from polysaccharides; protein tyrosine phosphatase 1B from glycoproteins; ganoderic acid A, C1, C2, and Df3 from triterpenoids; and ganoderol B can be cited as examples of bioactive compounds that exhibit antidiabetic and hypoglycemic mechanisms of action [71].

Li et al. [69], in their study examining the effects of *G. lucidum* polysaccharides and triterpenoids on cardiovascular diseases, applied a feeding program containing *G. lucidum* spore powder and oil at a dose of 0.3 g/kg bodyweight/day to rabbits for 4 months. The researchers reported a significant decrease in triglyceride and low-density lipoprotein cholesterol (LDL-C) levels in rabbits at the end of 4 months. It has also been reported that *G. lucidum* polysaccharides and triterpenoids inhibit the progression of atherosclerosis by reducing inflammatory polarization and the endothelial dysfunction of macrophages. It has also been proven in other studies that *G. lucidum* polysaccharides or extracts reduce serum glucose, insulin, lipid, triglyceride, and cholesterol levels [72,73]. Indeed, in a study investigating the effect of *G. lucidum* ethanol extract (50, 100, and 150 mg/kg) given to rats with diabetes, Tong et al. [73] found a positive effect of the extract and associated the improvement with free fatty acid metabolism. Moreover, Chen et al. [74] reported that the application of *G. lucidum* polysaccharides in mice at a dose of 400 mg/kg per day significantly reduced fasting blood glucose and insulin levels.

#### 1.2.5. The Mechanism of Action against COVID-19

COVID-19, announced by the World Health Organization (WHO) as a deadly epidemic worldwide on 11 March 2020, first entered our lives toward the end of 2019. The first case of the coronavirus disease originated in Wuhan, China. It was named severe acute respiratory syndrome coronavirus 2 (SARS-CoV-2) in 2003 due to symptoms similar to severe acute respiratory syndrome (SARS-CoV) [75,76,77]. The spread of this virus from Wuhan to 200 countries occurred in just 6 months [78]. As of 17 November 2022, the WHO reported a total of over 633,263,617 confirmed COVID-19 cases and over 6,594,491 deaths worldwide since the beginning of the pandemic [79]. COVID-19 can affect individuals differently, and the severity of the disease can depend on various factors, such as chronic disease status, immune system, age, smoking, and dietary habits [80]. Immunity in humans can be broadly classified into two types: innate immunity and acquired immunity. Innate immunity provides a general, nonspecific defense against a wide range of pathogens through physical and biochemical barriers and specialized immune cells, such as leukocytes (white blood cells); however, if the pathogen is able to breach these barriers and establish an infection, acquired immunity comes into play. Acquired immunity is characterized by the development of antigen-specific immune responses that can target and eliminate specific pathogens. Here, the reactions are carried out by T and B lymphocytes [78,81]. In this context, the adequate intake of macro- and micronutrients is essential for a strong and healthy immune system. Vitamins (such as vitamins A, B, C, D, and E) and minerals, such as iron and zinc [75,81,82,83], play important roles in immune function. Functional, encapsulated, and fortified foods are often used as supplements to strengthen immunity in COVID-19 infection [78]. Vitamin A plays a role in antibody production and can increase the levels of immunoglobulins G, M, and A. Vitamin B, specifically B6 and B12, is important for DNA production and repair and can also help with immune cell function. Vitamin C is a powerful antioxidant that can increase phagocytosis, oxidant production, and neutrophil migration to the site of infection. Vitamin D is important for the responsiveness of T cells, B cells, dendritic cells, and monocytes, which are all involved in immune function. Finally, vitamin E can increase lymphocyte proliferation, immunoglobulin levels, antibody responses, natural killer cell activity, interleukin-2 production, and macrophage activity, all of which can contribute to a stronger immune system. However, iron is essential for the production and maturation of immune cells, including lymphocytes. It is also involved in the production of cytokines, which are important for immune function. Zinc is involved in a wide range of immune functions, including the prevention of free radical-induced damage, the activation of macrophages and other immune cells, and the regulation of cytokine levels. Zinc is also important for the growth and differentiation of T and B cells, which are important for adaptive immunity [78,82,84,85].

While there is currently no specific treatment for COVID-19, research suggests that a healthy diet with the adequate intake of essential nutrients can help support the immune system and potentially reduce the severity of the disease. Several food components, such as vitamins, minerals, and phytochemicals, have been shown to have immunomodulatory properties and may play a role in fighting infectious agents such as COVID-19. In this context, *G. lucidum*, which has proven immunomodulatory and antiviral effects, has drawn attention. It has been determined that *G. lucidum* polysaccharides stimulate cellular and humoral immune reactions by increasing the production of T cells, cytokines (interleukin-2, interleukin 1β, interferon-y, and tumor necrosis factor-α), monocytes, and macrophages. Among these polysaccharides, β-glucans have an important role in the immune system through the dectin-1 receptor. It has been reported that β-glucans have a mechanism to prevent infections, including COVID-19, due to increased inflammatory responses by upregulating pattern recognition receptors. It is stated that molecular weight, branching, forms, and water solubility effect the physiological functions provided by β-glucans, while the ratio between the number of bonds and the side chain length and side branches effective their pharmacological properties.

Other compounds contributing to the antiviral activity of *G. lucidum* were determined to be ganoderic acids (GA-A, GA-B, GA-C1, GA-C2, GA-H, GA-T, GA-Q, and GA-β), ganodermanondiol, lucidumol, and ganodermanontriols. The antiviral activities of these compounds in *G. lucidum* have been proven against various viruses, such as coronavirus, hepatitis virus, human immunodeficiency virus (HIV), enterovirus, herpes simplex virus, H1N1 virus, and H5N1 virus. The unique immunomodulatory and antiviral properties of *G. lucidum* encourage its use as a therapeutic agent in the treatment of COVID-19 infection, which affects the whole world and causes many casualties [5,8,76]. On the other hand, in a study conducted, it was determined that phenolic compounds, such as luteolin and quercetin, might be helpful in the treatment of COVID-19, and the fact that *G. lucidum* contains these compounds can make it an important option in its treatment [86].

Based on the inhibitory properties of the bioactive components found in various fungi, such as *G. lucidum*, they have been reported to inhibit viruses’ entry into the host cell, intracellular adsorption, nucleic acid synthesis, viral replication, and ultimately viral proliferation. Furthermore, protease inhibitor activity is utilized in the development of antiviral drugs. In fact, it has been reported that *G. lucidum* can be used as a protease inhibitor in antiviral therapy [87]. In addition, beta-glucans in *G. lucidum* bind to macrophages, neutrophils, natural killer cells, and receptors on the outer membranes of cytotoxic T cells, triggering various reactions, activating white blood cells, and, thus, strengthening the immune system [88].

The study by AL-jumaili et al. [89] suggests that *G. lucidum* may have potential as a treatment for COVID-19. In this study, *G. lucidum* was given at a dose of 0.3 g/kg bodyweight to patients infected with COVID-19. They found that *G. lucidum* consumption was associated with an increase in positive immunoglobulin results, which may indicate an improvement in the body’s immune response to the virus. The study suggests that this immune-boosting effect may be due to the presence of various bioactive compounds in *G. lucidum*, including 3β-5α-dihydroxy-6-methoxyergosta-7,22-diene, ganolucidic acid β, ganoderic acid A–C, ganodermanontriol, ganodermanondiol, and lucidumol B. These compounds may have inhibitory effects against COVID-19 protease activity, which could potentially reduce viral replication and help to control the spread of the virus. In addition, the researchers explained the increase in white blood cells in patients consuming *G. lucidum* by the fact that the polysaccharides responsible for the immune activity modulation of *G. lucidum* support the stimulation of phagocytic activity and cytokine production, which act as mediators of inflammation. In addition, they reported that the bioactive terpenoids found in *G. lucidum* are widely used as antivirals to treat many diseases.

### 1.3. Potential Properties and Preparations of G. lucidum: A Need for Further Research and Quality Control

Notably, studies on the mechanisms of action of *G. lucidum* active constituents against metabolic disorders and coronavirus are not yet conclusive, and more research is needed to fully understand the potential properties of *G. lucidum*. Additionally, the active constituents and the mechanisms of action can vary depending on the source and extract methods of *G. lucidum* and the study design. There are many market preparations available that contain *G. lucidum* and related compounds. It is widely used as a traditional medicinal mushroom in many countries, and it has been incorporated into a variety of dietary supplements and herbal remedies. Some of the most common preparations of *G. lucidum* include the following:Capsules or tablets containing concentrated extracts of the mushroom.Powdered forms of the mushroom, which can be added to food or beverages.Tinctures or liquid extracts, which can be added to water or other liquids.Tea bags or loose tea leaves, which can be brewed similarly to traditional tea.

In addition to these preparations, there are also many other products on the market that contain compounds derived from *G. lucidum*, such as triterpenoids and polysaccharides. However, it is important to note that the quality and efficacy of these products can vary widely. Some preparations may contain little or no active compounds, while others may be contaminated with harmful substances [90].

### 1.4. Challenges and Limitations in the Clinical Application of G. lucidum Active Constituents against Metabolic Disorders and Coronavirus

There are several challenges and limitations to using *G. lucidum* active constituents in clinical applications against metabolic disorders and coronavirus. One of the main challenges is the lack of standardized products. There is currently no standardization regarding the type of *G. lucidum* used, the extraction method, or the composition of the active constituents. This lack of standardization makes it difficult to compare different studies’ results and determine the optimal dosage and administration method. Another challenge is the lack of large, well-designed clinical trials. Most of the studies on *G. lucidum* and its active constituents have been conducted in animals or cell cultures. The results of these studies may not be directly applicable to humans. Additionally, the available human studies are primarily observational or pilot studies with a small number of participants, which may not be strong enough to draw a definitive conclusion. There is also a lack of information about the safety of *G. lucidum* active constituents in the long term, especially in people with underlying medical conditions or those taking medication. There are also some concerns about the potential interactions between *G. lucidum* and other medications. Finally, it is important to note that *G. lucidum* and its active constituents have not yet been approved by regulatory agencies, such as the FDA, as a treatment for metabolic disorders or coronavirus, as more research is needed to confirm the safety and efficacy of these compounds.

## 2. Conclusions

*G. lucidum* is a mushroom that contains various bioactive compounds, including triterpenes, polysaccharides, peptides, and other compounds that have demonstrated potential pharmacological and therapeutic effects. Research has suggested that *G. lucidum* may have therapeutic effects on COVID-19 due to its antiviral and immunomodulatory properties. Some compounds found in *G. lucidum*, such as ganoderic acids, ganodermanondiol, lucidumol, and β-glucans, have shown potential antiviral effects and may help modulate the immune response to COVID-19. The mushroom also contains compounds that inhibit oxidation chain reactions, reduce lipid oxidation products, and suppress the activities of enzymes that produce toxic compounds, thereby reducing the damage caused by reactive oxygen species in our bodies and lipid oxidation in food processing. Furthermore, *G. lucidum* polysaccharides, glycoproteins, and triterpenes have shown anticancer activities through various mechanisms, including cytokine stimulation, cytotoxic effects, the inhibition of tumor cell adhesion, the blocking of apoptosis, and angiogenesis. *G. lucidum* extracts also exhibit beneficial effects in the treatment of cardiovascular diseases and diabetes; however, there are limitations to the mushroom’s utilization, including limited awareness of its potential and benefits outside of traditional Far Eastern medicine, limited cultivation regions, and the lack of widespread consumption. Further research on *G. lucidum* is necessary to understand its potential as a food supplement and in disease treatments.

## 3. Future Perspectives

Despite promising research on *G. lucidum* and its active components, it is important to recognize that further studies are required to fully comprehend their mechanisms of action and safety. It is also critical to acknowledge that self-medicating with *G. lucidum* supplements or extracts may not be advisable and could be hazardous for some individuals. As a result, it is always advisable to seek advice from a healthcare professional before taking any supplements.

## Figures and Tables

**Figure 1 foods-12-01512-f001:**
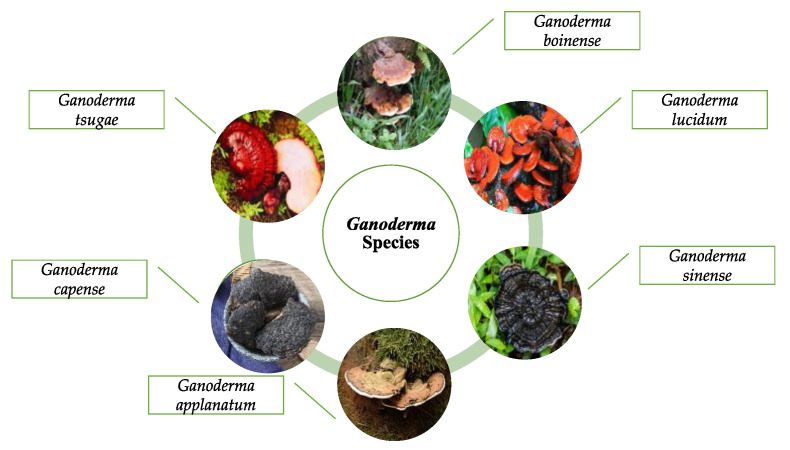
Images of some species of *Ganoderma*.

**Figure 2 foods-12-01512-f002:**
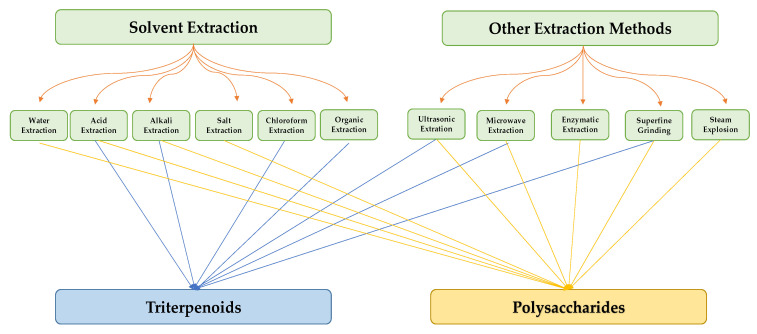
Extraction methods of triterpenoids and polysaccharides from *G. lucidum* [11,18].

**Figure 3 foods-12-01512-f003:**
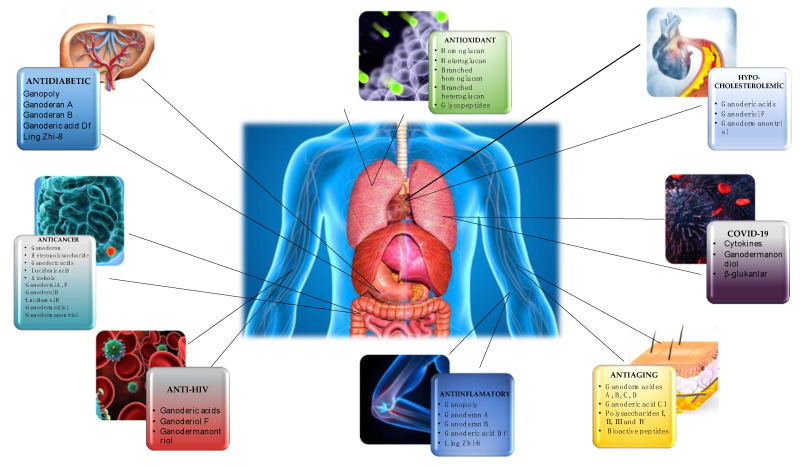
Therapeutic effects of compounds isolated from *G. lucidum* [4,8,10,15,23,26,27,28].

**Table 1 foods-12-01512-t001:** Different colors and applications of *G. lucidum*.

Color	Taste	Japanese Name	Usage Area	Reference
Blue	Sour	Aoshiba	Improves nervous system, eyesight and liver functions	Wasser [3]
Red	Bitter	Akashiba	Helps internal organs, improves memory, increases vitality	
Yellow	Sweet	Kishiba	Strengthens spleen function, calms the soul	
White	Bitter	Shiroshiba	Improves lung function, gives courage and strong will	
Black	Salty	Kuroshiba	Protects the kidneys	
Purple	Sweet	Murasakishiba	Improves the function of ears, joints, muscles, helps skin	

**Table 2 foods-12-01512-t002:** Anticancer mechanisms of compounds isolated from *G. lucidum* [15,23,47].

*G. lucidum* Compounds	Anticancer Mechanism of Action	Extract/Solvent/Extraction Method	Reference
Polysaccharides	○ Immune system stimulation (activating macrophages, natural killer cells, neutrophils, T-lymphocytes, B-lymphocytes, and cytotoxic T-lymphocytes, and inhibiting the growth of cytokines, such as TNF-α, IFN-c and IL-1β) ○ Cytotoxic effect ○ Apoptosis of tumor cells	Ethanol precipitation method/anion exchange chromatography	Lu et al. [23]; Sliva [47]
Ganoderic acid T 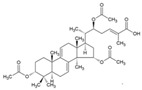	○ Urokinase plasminogen activator ○ Matrix metalloproteinase 2/9 (MMP2/9) ○ Arrest 95-D cell lines in the G1 cell cycle phase	Ethyl acetate/reversed phase high performance liquid chromatography (HPLC)	Ahmad [15]; Sliva [47]
Ganoderic acid DM 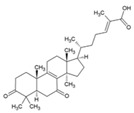	○ 5α-reductase inhibitory effect by modulating the androgen or estrogen receptor	Ethanol, methanol, ether, ethanol-water, and ethyl acetate	Ahmad [15]
Ganoderic acid C 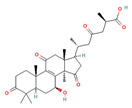	○ Competitive inhibition of protein prenyl transferase	Ethanol extracts	Ahmad [15]
Ganoderic acid H 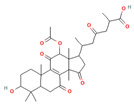	○ Reversing the abnormal behavior of activator protein 1 and nuclear factor-kappa B ○ Reducing urokinase-type plasminogen activator ○ Reducing expression of Cdk4		Ahmad [15]
Ganoderic acid A 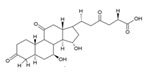	○ Reversing the abnormal behavior of activator protein 1 and nuclear factor-kappa B ○ Reducing urokinase-type plasminogen activator ○ Reducing expression of Cdk4 ○ Increasing chemosensitivity ○ Suppression of JAK-STAT-3 signaling helper proteins of JAK-1 and JAK-2		Ahmad [15]
Ganoderic acid y (GA-y) 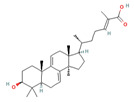 Ganoderic acid F (GA-F) 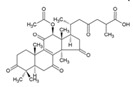 Ganoderic acid ε (GA-ε) 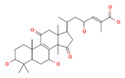	○ Inhibiting secondary metastatic growth		Ahmad [15]; Sliva [47]
Ganoderic acid Me 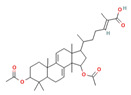	○ Antimetastasis ○ Inhibition of MMP9 and MMP2 gene expression ○ Inhibition of cell adhesion	Methanol extract	Ahmad [15] Wang et al. [50]
Lucidenic acid A 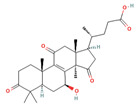 Lucidenic acid B 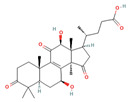 Lucidenic acid C 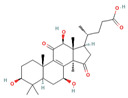 Lucidenic acid N 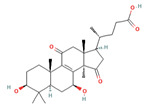	○ Apoptotic effect ○ Inhibiting the growth of cells	Reversed phase high performance liquid chromatography (HPLC)	Ahmad [15]; Sliva [47] Weng et al. [51]
Ganoderiol A 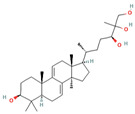 Ganoderiol B 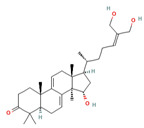 Ganoderiol F 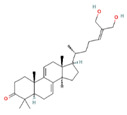 Lucidumol B 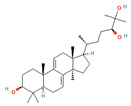 Ganodermatriol 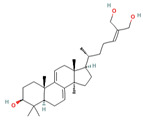 Ganodermanontriol (GDNT) 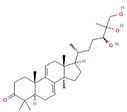	○ Antiproliferative properties ○ Cytotoxic effect	Methanol/HPLC	Ahmad [15]; Sliva [47] Min et al. [52]

## Data Availability

Not applicable.

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
