# Peer review of "Exploring the Potential Medicinal Benefits of Ganoderma lucidum: From Metabolic Disorders to Coronavirus Infections"

_foods, 2023, doi:10.3390/foods12071512_

Round 1

Reviewer 1 Report (Previous Reviewer 3)

Despite the paper has been improved, it is still far below the quality standards required by high IF journal. For instance, the main paragraphs of this article are lacking summary tables and, overall, the paper remain not well-organized as well as confusing.

Author Response

Response Letters to the Reviewers

Foods

foods-2238230

Title: Mechanisms of action of Ganoderma lucidum active constituents against metabolic disorders and coronavirus

Dear Respected Editor and Reviewers,

We sincerely thank the editor and all reviewers for their valuable feedback that we have used to improve the quality of our previous manuscript entitled “Mechanisms of action of Ganoderma lucidum active constituents against metabolic disorders and coronavirus” (ID: foods-2238230).

Thank you for handling our submission and offering us an opportunity for resubmission. We appreciate this hard-won opportunity very much. We have addressed all the comments and detailed all the changes made to the manuscript. We have studied the comments carefully and have made corrections, which are marked in red in the manuscript. We have tried our best to revise our manuscript based upon your concerns and comments. These changes will not influence the framework of the manuscript. Detailed information on the responses to the comments is presented as follows.

We earnestly appreciate the Editor/Reviewers’ work and hope that the corrections will be met with approval. I am looking forward to hearing from you at your convenience. We truly appreciate your help. Once again, thank you very much for giving us this opportunity to resubmit our manuscript. Many thanks for your help and have a lovely day.

Sincerely yours,

Prof. Dr. Fatih Oz

Ataturk University,

Food Engineering Department, 25240

Erzurum, Türkiye

Tel: +90-442-2312644

Response to the Reviewers' comments

Reviewer #1:

Comments and Suggestions for Authors

Despite the paper has been improved, it is still far below the quality standards required by high IF journal. For instance, the main paragraphs of this article are lacking summary tables and, overall, the paper remain not well-organized as well as confusing.

Thanks to the esteemed reviewer for their diligence and feedback. We are open to do further corrections, if needed, pending further clarification. Would you please clarify your specific concerns about the MS?. What specifically do you feel is lacking in the organization or presentation of the paper? Are there any specific elements you think should be added? We feel that your feedback can help to improve the overall quality and impact of your research.

Reviewer 2 Report (New Reviewer)

The manuscript entitled “Mechanisms of action of Ganoderma lucidum active constituents against metabolic disorders and coronavirus presents a review on the main active compouds from Ganoderma lucidum and their potential benefits in treating metabolic disorders such as diabetes and obesity, as well as its possible role in preventing and treating infections caused by the coronavirus.

The manuscript is very interesting and well-written. However, minor revisions should be made in order to be published in Foods journal, and the manuscript should be completed and/or modified taking into account the suggestions from the following comments.

The manuscript entitled “Mechanisms of action of Ganoderma lucidum active constituents against metabolic disorders and coronavirus” presents a review on the main active compouds from Ganoderma lucidum and their potential benefits in treating metabolic disorders such as diabetes and obesity, as well as its possible role in preventing and treating infections caused by the coronavirus. 
The manuscript is very interesting and well-written. However, minor revisions should be made, and the manuscript should be completed and/or modified taking into account the suggestions below: 
1. The authors are advised to rephrase the sentences from lines 38-40, 69-70, 82-83, 142-143, 194-196, 369-374, 448-452, 457-468, 471-472, 543-546, 569-571. 
2. The authors are advised to add the G. before naming the species, in order to have a complete Latin name (lines 85-86). For instance G. lucidum, G. applanatum, G. sinense, etc. 
3. The authors are advised to explain about which Ganoderma species they write (see lines 156, 170, 171, etc). Ganoderma is the name of plant genus, especially if the Italic style is used. 
4. From the fig. 3, one can understand that cytokines are isolated compounds from G. lucidum, please correct. 
5. The authors are advised to complete the subsections 1.2.1, 1.2.2, 1.2.3, 1.2.5 with details about the extracts and doses used in each study 
6. The authors are advised to use italic style for p (p-coumaric acid – line 247). 
7. In table 2, the authors are advised to complete for each class of compounds the references, doses and extracts used. 
8. Taking into account the paragraph from lines 515-519, the presentation of details about the extracts and doses used can be important, therefore thee authors are advised to complete the manuscript.

Author Response

Response Letters to the Reviewers

Foods

foods-2238230

Title: Mechanisms of action of Ganoderma lucidum active constituents against metabolic disorders and coronavirus

Dear Respected Editor and Reviewers,

We sincerely thank the editor and all reviewers for their valuable feedback that we have used to improve the quality of our previous manuscript entitled “Mechanisms of action of Ganoderma lucidum active constituents against metabolic disorders and coronavirus” (ID: foods-2238230).

Thank you for handling our submission and offering us an opportunity for resubmission. We appreciate this hard-won opportunity very much. We have addressed all the comments and detailed all the changes made to the manuscript. We have studied the comments carefully and have made corrections, which are marked in red in the manuscript. We have tried our best to revise our manuscript based upon your concerns and comments. These changes will not influence the framework of the manuscript. Detailed information on the responses to the comments is presented as follows.

We earnestly appreciate the Editor/Reviewers’ work and hope that the corrections will be met with approval. I am looking forward to hearing from you at your convenience. We truly appreciate your help. Once again, thank you very much for giving us this opportunity to resubmit our manuscript. Many thanks for your help and have a lovely day.

Sincerely yours,

Prof. Dr. Fatih Oz

Ataturk University,

Food Engineering Department, 25240

Erzurum, Türkiye

Tel: +90-442-2312644

Response to the Reviewers' comments

Reviewer #2:

Comments and Suggestions for Authors

The manuscript entitled “Mechanisms of action of Ganoderma lucidum active constituents against metabolic disorders and coronavirus” presents a review on the main active compouds from Ganoderma lucidum and their potential benefits in treating metabolic disorders such as diabetes and obesity, as well as its possible role in preventing and treating infections caused by the coronavirus.

The manuscript is very interesting and well-written. However, minor revisions should be made in order to be published in Foods journal, and the manuscript should be completed and/or modified taking into account the suggestions from the following comments.

The manuscript entitled “Mechanisms of action of Ganoderma lucidum active constituents against metabolic disorders and coronavirus” presents a review on the main active compouds from Ganoderma lucidum and their potential benefits in treating metabolic disorders such as diabetes and obesity, as well as its possible role in preventing and treating infections caused by the coronavirus.

The manuscript is very interesting and well-written. However, minor revisions should be made, and the manuscript should be completed and/or modified taking into account the suggestions below:

Thank you very much for giving us this opportunity and your valuable comments. We have carefully revised the manuscript. We hope that the revised article has met the publication criteria of yours.

  1. The authors are advised to rephrase the sentences from lines 38-40, 69-70, 82-83, 142-143, 194-196, 369-374, 448-452, 457-468, 471-472, 543-546, 569-571.

Thank you for your comment and caution. The related sentences have been rephrased. Please go through the revised manuscript.

  1. The authors are advised to add the G. before naming the species, in order to have a complete Latin name (lines 85-86). For instance G. lucidum, G. applanatum, G. sinense, etc.

Thank you for your comment and caution. This part has been revised. Please see the line of 89 in the untracked revised manuscript.

  1. The authors are advised to explain about which Ganoderma species they write (see lines 156, 170, 171, etc). Ganoderma is the name of plant genus, especially if the Italic style is used.

Thank you for your comment. It is true. The name of the genus should be written in italics.

  1. From the fig. 3, one can understand that cytokines are isolated compounds from G. lucidum, please correct.

Thank you for your comment. While Ganoderma lucidum contains a variety of bioactive compounds, including polysaccharides, triterpenes, and sterols, cytokines are not typically considered to be one of the major active compounds in this mushroom. However, some studies have suggested that consumption of Ganoderma lucidum extracts may have an impact on cytokine production and immune function in the body.

  1. The authors are advised to complete the subsections 1.2.1, 1.2.2, 1.2.3, 1.2.5 with details about the extracts and doses used in each study

Thank you for your comment. The related information has been added to the subsections you mentioned. Please go through the subsections.

  1. The authors are advised to use italic style for p (p-coumaric acid – line 247).

Thank you for your comment. The related correction has been done. Please see the line of 255 in the untracked revised manuscript.

  1. In table 2, the authors are advised to complete for each class of compounds the references, doses and extracts used.

Thank you for your comment. The related information has been added to Table 2. Please go through Table 2.

  1. Taking into account the paragraph from lines 515-519, the presentation of details about the extracts and doses used can be important, therefore thee authors are advised to complete the manuscript.

Thank you for your comment. The information (extraction technique and usage dose) given in the articles cited in the relevant sections has been given in the revised article. Please go through the revised manuscript.

Reviewer 3 Report (New Reviewer)

The manuscript entitled “Mechanisms of action of Ganoderma lucidum active constituents against metabolic disorders and coronavirus” summarizes the potential benefits of G. lucidum in treating metabolic disorders such as diabetes and obesity, as well as its possible role in preventing and treating infections caused by the coronavirus. The authors discussed that the triterpenoids in Ganoderma lucidum have demonstrated anti-inflammatory and antioxidant properties. These compounds have been found to improve insulin sensitivity and lower blood sugar levels in animal models of diabetes. Additionally, G. lucidum polysaccharides have been found to reduce body weight and improve glucose metabolism in animal models of obesity. These polysaccharides can also help to increase the activity of specific white blood cells, which play a critical role in the body's immune response. For coronavirus, some in vitro studies have suggested that G. lucidum polysaccharides and triterpenoids have an inhibitory effect on the growth of coronaviruses. However, these results have not been verified through clinical trials, so it would be premature to draw any definitive conclusions about the effectiveness of G. lucidum in preventing or treating coronavirus infections in humans.

 The below revisions are recommended:

  1. Abstract (Lines # 32-34): “One of the main active constituents of G. lucidum is a group of compounds called triterpenoids……….” Rephrase the sentence.
  2. Lines 107-113: Cite appropriate references based on which the components of G. lucidum were listed I I, II, and III.
  3. Figure 3: Under the subtitle “Anticancer’ mention the name of the ‘Alcohols.’
  4. Table 2: Draw the chemical structures of the small molecules isolated from G. lucidum

  1. Based on the ‘Author’s opinion against literature’ (Line 514) and ‘Challenges and limitations in the clinical application of G. lucidum active constituents against metabolic disorders and coronavirus’ (Lines 520-521), the title of the article is confusing, and it must be changed so that the title reflects the actual findings.

  1. The manuscript must be thoroughly checked, and the quality of the language must be improved. There are numerous grammatical mistakes.

  1. Uniformity (font and size) should be mentioned throughout the manuscript, including the schemes and figures. The authors are encouraged to check the journal IFA.

Author Response

Response Letters to the Reviewers

Foods

foods-2238230

Title: Mechanisms of action of Ganoderma lucidum active constituents against metabolic disorders and coronavirus

Dear Respected Editor and Reviewers,

We sincerely thank the editor and all reviewers for their valuable feedback that we have used to improve the quality of our previous manuscript entitled “Mechanisms of action of Ganoderma lucidum active constituents against metabolic disorders and coronavirus” (ID: foods-2238230).

Thank you for handling our submission and offering us an opportunity for resubmission. We appreciate this hard-won opportunity very much. We have addressed all the comments and detailed all the changes made to the manuscript. We have studied the comments carefully and have made corrections, which are marked in red in the manuscript. We have tried our best to revise our manuscript based upon your concerns and comments. These changes will not influence the framework of the manuscript. Detailed information on the responses to the comments is presented as follows.

We earnestly appreciate the Editor/Reviewers’ work and hope that the corrections will be met with approval. I am looking forward to hearing from you at your convenience. We truly appreciate your help. Once again, thank you very much for giving us this opportunity to resubmit our manuscript. Many thanks for your help and have a lovely day.

Sincerely yours,

Prof. Dr. Fatih Oz

Ataturk University,

Food Engineering Department, 25240

Erzurum, Türkiye

Tel: +90-442-2312644

Response to the Reviewers' comments

Reviewer #3:

Comments and Suggestions for Authors

The manuscript entitled “Mechanisms of action of Ganoderma lucidum active constituents against metabolic disorders and coronavirus” summarizes the potential benefits of G. lucidum in treating metabolic disorders such as diabetes and obesity, as well as its possible role in preventing and treating infections caused by the coronavirus. The authors discussed that the triterpenoids in Ganoderma lucidum have demonstrated anti-inflammatory and antioxidant properties. These compounds have been found to improve insulin sensitivity and lower blood sugar levels in animal models of diabetes. Additionally, G. lucidum polysaccharides have been found to reduce body weight and improve glucose metabolism in animal models of obesity. These polysaccharides can also help to increase the activity of specific white blood cells, which play a critical role in the body's immune response. For coronavirus, some in vitro studies have suggested that G. lucidum polysaccharides and triterpenoids have an inhibitory effect on the growth of coronaviruses. However, these results have not been verified through clinical trials, so it would be premature to draw any definitive conclusions about the effectiveness of G. lucidum in preventing or treating coronavirus infections in humans.

Thank you very much for giving us this opportunity and your valuable comments. We have carefully revised the manuscript. We hope that the revised article has met the publication criteria of yours.

The below revisions are recommended:

  1. Abstract (Lines # 32-34): “One of the main active constituents of G. lucidum is a group of compounds called triterpenoids……….” Rephrase the sentence.

Thank you for your comment. The related sentence has been rephrased. Please see the lines of 32 and 34 in the untracked revised manuscript.

  1. Lines 107-113: Cite appropriate references based on which the components of G. lucidum were listed I I, II, and III.

Thank you for your comment. The appropriate references have been cited in this section. Please see the lines of 115 and 121 in the untracked revised manuscript.

  1. Figure 3: Under the subtitle “Anticancer’ mention the name of the ‘Alcohols.’

Thank you for your comment. The names of the “alcohols” have been mentioned under the subtitle “Anticancer” in Figure 3. Please go through the Figure 3.

  1. Table 2: Draw the chemical structures of the small molecules isolated from G. lucidum

Thank you for your comment. The chemical structures of the small molecules isolated from G. lucidum have been added to Table 2. Please go through the Table 2.

  1. Based on the ‘Author’s opinion against literature’ (Line 514) and ‘Challenges and limitations in the clinical application of G. lucidum active constituents against metabolic disorders and coronavirus’ (Lines 520-521), the title of the article is confusing, and it must be changed so that the title reflects the actual findings.

Thank you for your comment. In line with your suggestion, the title of the manuscript has been changed as “Exploring the potential medicinal benefits of Ganoderma lucidum: from metabolic disorders to coronavirus infections”. Please see the title of the revised manuscript.

  1. The manuscript must be thoroughly checked, and the quality of the language must be improved. There are numerous grammatical mistakes.

Thank you for your comment. The manuscript has been thoroughly checked and the quality of the language has been improved. Grammatical mistakes have been fixed. Please go through the revised manuscript.

  1. Uniformity (font and size) should be mentioned throughout the manuscript, including the schemes and figures. The authors are encouraged to check the journal IFA.

Thank you for your comment. Uniformity has been ensured in terms of writing in the article. Please go through the revised manuscript. Please go through the revised manuscript.

Reviewer 4 Report (New Reviewer)

The Review entitled  ''Mechanisms of action of Ganoderma lucidum active constituents against metabolic disorders and coronavirus'' is interesting and organised. However, the authors need to make some changes.

-        Please, cite a relevant reference in Tables 1 and 2 in an additional column.

-        In author's opinion against literature, please add more explanations and support your opinion with different examples of previous literature.

-        I wonder if there are market preparations for Ganoderma lucidum and related compounds. The author should discuss this point.

-        Authors need to correct some grammatical mistakes.

-        Please, mention this sentence only once at the end of the manuscript.

''The image was created by the group.''

-        Please check the abbreviations throughout the manuscript. It would be best to introduce the acronym when the whole word appears the first time in the text and then use only the abbreviations.

Author Response

Response Letters to the Reviewers

Foods

foods-2238230

Title: Mechanisms of action of Ganoderma lucidum active constituents against metabolic disorders and coronavirus

Dear Respected Editor and Reviewers,

We sincerely thank the editor and all reviewers for their valuable feedback that we have used to improve the quality of our previous manuscript entitled “Mechanisms of action of Ganoderma lucidum active constituents against metabolic disorders and coronavirus” (ID: foods-2238230).

Thank you for handling our submission and offering us an opportunity for resubmission. We appreciate this hard-won opportunity very much. We have addressed all the comments and detailed all the changes made to the manuscript. We have studied the comments carefully and have made corrections, which are marked in red in the manuscript. We have tried our best to revise our manuscript based upon your concerns and comments. These changes will not influence the framework of the manuscript. Detailed information on the responses to the comments is presented as follows.

We earnestly appreciate the Editor/Reviewers’ work and hope that the corrections will be met with approval. I am looking forward to hearing from you at your convenience. We truly appreciate your help. Once again, thank you very much for giving us this opportunity to resubmit our manuscript. Many thanks for your help and have a lovely day.

Sincerely yours,

Prof. Dr. Fatih Oz

Ataturk University,

Food Engineering Department, 25240

Erzurum, Türkiye

Tel: +90-442-2312644

Response to the Reviewers' comments

Reviewer #4:

Comments and Suggestions for Authors

The Review entitled ''Mechanisms of action of Ganoderma lucidum active constituents against metabolic disorders and coronavirus'' is interesting and organised. However, the authors need to make some changes.

Thank you very much for giving us this opportunity and your valuable comments. We have carefully revised the manuscript. We hope that the revised article has met the publication criteria of yours.

-       Please, cite a relevant reference in Tables 1 and 2 in an additional column.

Thank you for your comment. The related references have been added to Tables 1 and 2 as an additional column. Please go through the Table 1 and Table 2.

-       In author's opinion against literature, please add more explanations and support your opinion with different examples of previous literature.

Thank you for your comment. Our opinion came after reviewing the available literature and it came in short. This is a general conclusion of our team on the subject.

-       I wonder if there are market preparations for Ganoderma lucidum and related compounds. The author should discuss this point.

Thank you for your comment. Yes, there are many market preparations available that contain Ganoderma lucidum and related compounds. It is widely used as a traditional medicinal mushroom in many countries, and it has been incorporated into a variety of dietary supplements and herbal remedies. Some of the most common preparations of Ganoderma lucidum include:

1- Capsules or tablets containing concentrated extracts of the mushroom.

2- Powdered forms of the mushroom, which can be added to food or beverages.

3- Tinctures or liquid extracts, which can be added to water or other liquids.

4- Tea bags or loose tea leaves, which can be brewed like traditional tea.

In addition to these preparations, there are also many other products on the market that contain compounds derived from Ganoderma lucidum, such as triterpenoids and polysaccharides. It is important to note, however, that the quality and efficacy of these products can vary widely. Some preparations may contain little or no active compounds, while others may be contaminated with harmful substances. These information has been added to the text. Please see the lines of 669 and 682 in the untracked revised manuscript.

-       Authors need to correct some grammatical mistakes.

Thank you for your comment. The quality of the language of the manuscript has been improved. Please go through the revised manuscript.

-       Please, mention this sentence only once at the end of the manuscript.

''The image was created by the group.''

Thank you for your comment. The related information has been moved to the end of the revised manuscript and the sentence has been removed from the Figures’ captions. Please see the lines of 92, 207, and 653 and 654.

-       Please check the abbreviations throughout the manuscript. It would be best to introduce the acronym when the whole word appears the first time in the text and then use only the abbreviations.

Thank you for your comment. The related corrections have been done in the revised manuscript. Please go through the revised manuscript.

Round 2

Reviewer 1 Report (Previous Reviewer 3)

In my humble opinion, despite the manuscript has been improved, it is still far below the quality standards required by high IF journal.

Author Response

Response Letters to the Reviewers

Foods

foods-2238230

Title: Mechanisms of action of Ganoderma lucidum active constituents against metabolic disorders and coronavirus

Reviewer #1:

Comments and Suggestions for Authors

In my humble opinion, despite the manuscript has been improved, it is still far below the quality standards required by high IF journal.

Thank you for taking the time to review our manuscript and for your valuable feedback. We appreciate your honest opinion and are grateful for your input. We are pleased to hear that you found the manuscript improved since its initial submission. However, we are sorry that you still believe it falls below the quality standards required by high-impact factor journals. We would be grateful if you could provide us with more specific feedback on areas that need further improvement. This would allow us to address any remaining issues and improve the manuscript to meet the standards expected by the journal. Please be informed that other reviewers and the editor have provided positive feedback that supports the manuscript's publication.

Reviewer 3 Report (New Reviewer)

There are still some grammatical mistakes. I recommend minor revision before publication.

Author Response

Response Letters to the Reviewers

Foods

foods-2238230

Title: Mechanisms of action of Ganoderma lucidum active constituents against metabolic disorders and coronavirus

Dear Respected Editor and Reviewers,

We sincerely thank the editor and all reviewers for their valuable feedback that we have used to improve the quality of our previous manuscript entitled “Mechanisms of action of Ganoderma lucidum active constituents against metabolic disorders and coronavirus” (ID: foods-2238230).

Thank you for handling our submission and offering us an opportunity for resubmission. We appreciate this hard-won opportunity very much. We have addressed all the comments and detailed all the changes made to the manuscript. We have studied the comments carefully and have made corrections, which are marked in red in the manuscript. We have tried our best to revise our manuscript based upon your concerns and comments. These changes will not influence the framework of the manuscript. Detailed information on the responses to the comments is presented as follows.

We earnestly appreciate the Editor/Reviewers’ work and hope that the corrections will be met with approval. I am looking forward to hearing from you at your convenience. We truly appreciate your help. Once again, thank you very much for giving us this opportunity to resubmit our manuscript. Many thanks for your help and have a lovely day.

Sincerely yours,

Prof. Dr. Fatih Oz

Ataturk University,

Food Engineering Department, 25240

Erzurum, Türkiye

Tel: +90-442-2312644

Response to the Reviewers' comments

Reviewer #3:

Comments and Suggestions for Authors

There are still some grammatical mistakes. I recommend minor revision before publication.

Thank you for taking the time to review our work and providing valuable feedback. We appreciate your attention and have improved the grammatical accuracy throughout the text.

Reviewer 4 Report (New Reviewer)

The authors revised the manuscript according to the recommendations. 

I have only one comment, please remove the title '' Author’s opinion against literature'' or change it to another one to be more convenient with the paragraph below.

Author Response

Response Letters to the Reviewers

Foods

foods-2238230

Title: Mechanisms of action of Ganoderma lucidum active constituents against metabolic disorders and coronavirus

Dear Respected Editor and Reviewers,

We sincerely thank the editor and all reviewers for their valuable feedback that we have used to improve the quality of our previous manuscript entitled “Mechanisms of action of Ganoderma lucidum active constituents against metabolic disorders and coronavirus” (ID: foods-2238230).

Thank you for handling our submission and offering us an opportunity for resubmission. We appreciate this hard-won opportunity very much. We have addressed all the comments and detailed all the changes made to the manuscript. We have studied the comments carefully and have made corrections, which are marked in red in the manuscript. We have tried our best to revise our manuscript based upon your concerns and comments. These changes will not influence the framework of the manuscript. Detailed information on the responses to the comments is presented as follows.

We earnestly appreciate the Editor/Reviewers’ work and hope that the corrections will be met with approval. I am looking forward to hearing from you at your convenience. We truly appreciate your help. Once again, thank you very much for giving us this opportunity to resubmit our manuscript. Many thanks for your help and have a lovely day.

Sincerely yours,

Prof. Dr. Fatih Oz

Ataturk University,

Food Engineering Department, 25240

Erzurum, Türkiye

Tel: +90-442-2312644

Response to the Reviewers' comments

Reviewer #4:

Comments and Suggestions for Authors

The authors revised the manuscript according to the recommendations.

I have only one comment, please remove the title '' Author’s opinion against literature'' or change it to another one to be more convenient with the paragraph below.

Thank you. We have amended the heading as requested.

This manuscript is a resubmission of an earlier submission. The following is a list of the peer review reports and author responses from that submission.

Round 1

Reviewer 1 Report

The scope of the review is interesting but lacks novelty as the medicinal values of Ganoderma lucidum have been extensively reviewed. Moreover, the taxonomy of the species called G. lucidum is being debated. Many information has been compiled but there is insufficient critical analysis of the information presented. The abstract should be rewritten to reflect the content of this manuscript.

Reviewer 2 Report

Abstract: Highlight the mechanism of active metabolite against metabolic disorders and COVID in abstract.

If this mushroom is not edible, please explain the significance of it with relation to food and what possible application in food can be demonstrated.

Table 1 can be described in the text and remove it as a table. 

Please write all scientific names in italics, eq. line 79.

Figure 1: please change the caption from some types to appropriate word.

Please describe the mechanism of action against the COVID with some valid references, as review claims the mechanism of action, but general discussion and supportive data do not justify the claim made. Enlist the bioactive constituents and their mechanism of action against metabolic disorders and COVID-19. 

Same concern is observed in other mechanisms described, rather than general discussion, please be specific and provide the mechanism of action of active principles of the mushroom under consideration.

Rewrite the conclusion and be specific with the findings rather than repeating the information from abstract and result section in conclusion. 

Which data basis were considered for data collection, how many previous research studies are available and considered for the review??

Reviewer 3 Report

The paper “Mechanisms of action of Ganoderma lucidum active constituents against metabolic disorders and coronavirus” by Elif Ekiz and co-workers reviewed the Ganoderma lucidum active constituents in the management of metabolic disorders and coronavirus, focusing on its mechanism of action.

The discussed subject is interesting because nutraceuticals can play a role in the treatment of both metabolic disorders and coronavirus. Nevertheless, this paper is dispersive. In addition, the article is not well set-up and in almost each paragraph is reported a very long introduction, failing into highlight the major points of the study. Moreover, the main paragraphs of this article are lacking summary tables, while it is reported in the “anticancer activity mechanisms” paragraph. Therefore, I suggest Authors to reorganize the paper focusing on the main topic of the review.

Reviewer 4 Report

The current review about the mechanisms of action of bioactive compounds from Ganoderma on metabolic disorders and COVID, is novel and very interesting. Further, the manuscript is very well organised and easily readable.

Line 79. “G. lucidum” must be in italics 

Line 80. Missing the year/number of the cite “Gen.”

Section “1.2.1. The Mechanism of Impact on COVID-19” must be moved after “1.2.5. Antidiabetes Activity Mechanism”

Round 2

Reviewer 1 Report

The authors have addressed my comments. There are two things that still require their attention.

1. The authors misinterpreted my comment on the taxonomy of this mushroom species. What I meant was there is a dispute on the correct scientific name for the species that was formerly known as Ganoderma lucidum. In a number of publications, several authors have attempted to revise the scientific name for this species. This should be reflected in the review.

2. If the images in Fig 3 are obtained elsewhere, permission maybe needed to reproduce them in this article.

Reviewer 2 Report

The authors have revised the manuscript as advised, however, the mechanism of action against COVID is still not concrete and based on assumptions or previously proposed findings. 

Reviewer 3 Report

The Authors have not addressed any of the points I discussed in my first revision. Moreover, the additions they claim to have done, probably referred to the revisions of the other reviewers, do not fill the gaps I highlighted.